# Addressing Malnutrition Through Reducing the Cost of a Healthy Diet in Bangladesh

**DOI:** 10.3390/foods14244237

**Published:** 2025-12-10

**Authors:** Nazma Shaheen, Abira Nowar, Saiful Islam, Md. Hafizul Islam, Mohammad Monirul Hasan, Rudaba Khondker, Zoe Odette Barois, Just Dengerink

**Affiliations:** 1Institute of Nutrition and Food Science, University of Dhaka, Dhaka 1000, Bangladesh; abira.infs@du.ac.bd (A.N.); saifulinfs@du.ac.bd (S.I.); hafizinfs19@gmail.com (M.H.I.); 2Global Alliance for Improved Nutrition (GAIN), Dhaka 1212, Bangladesh; mmonirul@gainhealth.org (M.M.H.); rkhondker@gainhealth.org (R.K.); 3Wageningen Social & Economic Research, Wageningen University & Research, Droevendaalsesteeg 4, 6708 PB Wageningen, The Netherlands; zoe.barois@wur.nl (Z.O.B.); just.dengerink@wur.nl (J.D.)

**Keywords:** food system transformation, foresight analysis, malnutrition, healthy diet, cost of healthy diet, Bangladesh

## Abstract

Bangladesh has significantly reduced child undernutrition, yet micronutrient deficiencies and diet-related non-communicable diseases remain pressing challenges. While the afordability of healthy diets is recognized as a key determinant of nutrition outcomes, limited attention has been paid to the uncertainties that affect diet costs and access over time. This paper addresses this gap by exploring the major drivers of uncertainty in the cost of healthy diets in Bangladesh and their implications for nutrition policy. This study emloyed foresight tools to explore future uncertainties affecting the cost and accessibility of healthy diets in Bangladesh. Key drivers of change, such as climate variability, market dynamics, income inequality, and dietary behavior, were identified through a structured expert workshop. Two critical uncertainties were selected using the 2 × 2 scenario planning method: food price volatility and changing dietary patterns. These formed the basis for four plausible future scenarios, each illustrating different trajectories for nutrition and food system outcomes. This foresight approach supports proactive, multisectoral policymaking by highlighting potential risks and opportunities for ensuring affordable, nutritious diets in a changing context. The resulting scenarios underscore the need for integrated, multisectoral strategies to build resilient food systems, improve the affordability of nutrient-rich foods, and promote dietary behavior change.

## 1. Introduction

Malnutrition has long been a major public health challenge in Bangladesh. Since its independence in 1971, the country has made remarkable progress in reducing maternal and child undernutrition and achieving the Millennium Development Goals (MDGs), while advancing towards the Sustainable Development Goals (SDGs) and global initiatives such as the UN Food System Summit and Nutrition for Growth. National surveys illustrate this progress. The prevalence of stunting among children under five fell from 60% in 1997 to 23.6% in 2022, while women’s chronic energy deficiency declined from 52% in 1997 to 8.9% in 2022 [1]. Public health interventions such as vitamin A supplementation, salt iodization, and food fortification significantly reduced micronutrient deficiencies [2,3,4,5,6,7]. However, the nutrition transition has given rise to new challenges: the prevalence of underweight and wasting among children has increased since 2017 [1,8]. Overweight and obesity among women of reproductive age have risen sharply, now affecting more than one-third of women [1]. In parallel, diet- and lifestyle-related non-communicable diseases (NCDs) such as hypertension and diabetes are on the rise, underscoring the double burden of malnutrition in Bangladesh [4,9].

Food affordability plays a critical role in this context. According to UNICEF’s conceptual framework of malnutrition, access to affordable, safe, and nutritious foods is an essential determinant of child and maternal nutrition. Despite progress in poverty reduction—from 34.3% in 2000 to 18.7% in 2022—many households remain food insecure, with 30.5% experiencing moderate or severe food insecurity in 2021–2023 [10,11]. The rising cost of nutritious foods such as fruits, vegetables, pulses, and animal-source foods poses a significant barrier to adopting recommended diets, particularly for lower-income households [12]. Recent estimates suggest that in 2022 the cost of a healthy diet in Bangladesh was USD 3.20 per person per day, a figure that remains unaffordable for a significant proportion of the population, even though the share unable to afford a healthy diet has declined from 75% in 2017 to 66% in 2021 [13]. These figures indicate that, while progress has been made, affordability remains a persistent obstacle to achieving adequate nutrition for all. Addressing this issue requires not only income growth and poverty reduction, but also food system interventions to improve the accessibility and affordability of nutrient-rich foods.

A healthy diet, as defined in the national Food-Based Dietary Guidelines (FBDG), emphasizes proportional intake from eight food groups are cereals; pulses; vegetables; fruits; meat, fish, and eggs; milk and dairy products; fats and oils; and sugar [14]. While consumption of fruits and vegetables has increased over the past decade, average intake remains well below the WHO-recommended 400 g per day [15]. Intake disparities persist by income level, with lower-income households consuming significantly less, largely due to affordability barriers [16,17,18]. Moreover, food price volatility, urban–rural differences, and seasonal fluctuations further exacerbate these disparities. This gap between dietary recommendations and actual intake not only undermines progress in addressing undernutrition but also contributes to the rising burden of diet-related NCDs.

Taken together, these patterns highlight the central role of food affordability in shaping dietary intake and nutritional outcomes in Bangladesh. Despite notable achievements in reducing malnutrition over the past decades, progress is increasingly constrained by the high and rising cost of a healthy diet, which limits access for vulnerable groups and slows progress towards the SDGs. Understanding how the price of a healthy diet influences dietary intake is therefore critical for designing effective policies and interventions. This study aims to provide foresight on the cost and intake scenarios of a healthy diet in Bangladesh, and to explore how reducing diet costs could contribute to addressing malnutrition, with particular emphasis on policy relevance for ensuring equitable access to healthy diets.

## 2. Materials and Methods

### 2.1. Study Design

This policy paper employed a mixed-methods approach, combining secondary data analysis, literature review, and expert consultation within the Foresight for Food framework to explore key uncertainties and future pathways related to the cost of healthy diets in Bangladesh. The foresight approach provided a structured process for integrating evidence and expert judgment to address the complexity of the national food system and its transformation challenges. The process began with system scoping and mapping to define the boundaries of the food system, identify key actors, and visualize interactions among drivers such as food prices, dietary patterns, urbanization, and policy coherence. Insights derived from literature and expert workshops were used to identify and prioritize critical trends and uncertainties shaping access to affordable, healthy diets. Scenario analysis was then applied to explore how these uncertainties might evolve and to assess their potential consequences for nutrition and food affordability. Through participatory foresight exercises, multiple plausible future scenarios were developed, reflecting diverse policy and socioeconomic trajectories. An approach was subsequently employed to identify leverage points and strategic actions required to transition from the current state toward more equitable and sustainable food system futures.

### 2.2. Data Sources

Household income, expenditure, and food consumption data were obtained from the Household Income and Expenditure Survey (HIES) [11]. Food price data for key staples, fruits, vegetables, pulses, and animal-source foods were obtained from FAOSTAT [13] and national market surveys. Nutritional requirements and recommended intake patterns were based on the Food-Based Dietary Guidelines (FBDG) for Bangladesh [14]. Additional literature and reports were accessed to identify factors influencing diet cost and malnutrition trends [10,13,19,20,21,22,23,24,25,26,27,28,29,30,31,32,33,34].

### 2.3. Definition of a Healthy Diet

A healthy diet was defined according to the national FBDG, which emphasizes proportional intake from eight food groups: cereals; pulses; vegetables; fruits; meat, fish, and eggs; milk and milk products; fats and oils; and sugar [14]. The diet aims to meet nutrient adequacy, support overall health, and prevent malnutrition, including undernutrition and diet-related non-communicable diseases (NCDs). Serving sizes and recommended ranges from the FBDG were used to construct the diet for a typical adult, forming the basis for cost calculations.

### 2.4. Identification and Selection of Key Uncertainties

Key uncertainties affecting the cost of a healthy diet were identified using a structured approach:

Literature Review: A literature search was conducted for key documents (peer-reviewed articles, reports, and policy documents) on food affordability, cost of healthy diets, nutrition, and food system challenges in Bangladesh. Priority was given to studies published in the past 10 years and those reporting quantitative or qualitative evidence on drivers of diet cost.

Expert Consultation: Three national workshops were convened in Dhaka, including 60–70 stakeholders representing policymakers, nutritionists, agricultural experts, market analysts, program implementers, and public health professionals. The list of participants has been presented in the Appendix A. Experts were selected based on their experience in nutrition, food security, agricultural economics, and policy implementation. Selection criteria included prior research publications, active engagement in national programs, and professional affiliation with government agencies, universities, or NGOs.

Selection Process: Experts reviewed a preliminary list of potential uncertainties from the literature and provided additional inputs based on practical experiences. They rated each uncertainty on two dimensions: degree of uncertainty (likelihood of change) and degree of significance (impact on CoHD and diet affordability). Based on this rating, two critical uncertainties, food price volatility and dietary pattern changes, were selected for scenario development.

### 2.5. Scenario Development and Establishing Implications

The two selected uncertainties were combined in a 2 × 2 matrix to generate four plausible scenarios of healthy diet cost and dietary intake: (1) Optimal Health for All: Affordable, nutrient-rich diets; (2) High Price for Healthy Eating: High-cost, healthy diets; (3) Crisis of Nutrition: Unaffordable, unhealthy diets; (4) Unhealthy Eating Despite Affordable Food Prices: Low-cost but nutritionally poor diets.

Role of Experts on establishing the implications of the scenarios: Experts participated in structured discussions to evaluate each scenario and identify potential impact on food system actors (consumers, farmers, retailers, policymakers). Consensus-derived implications, considering how price changes and dietary patterns might influence production, consumption, market dynamics, and policy interventions. This participatory approach ensured that scenarios reflected empirical evidence and key stakeholders’ contextual knowledge.

### 2.6. Ethical Considerations

Ethical approval was not required because the study was based on publicly available secondary data and expert consultation. All data were used in aggregate form, ensuring no individual-level identifiers were included.

## 3. Results

### 3.1. Key Uncertainties Affecting the Cost of a Healthy Diet in Bangladesh

Several key uncertainties of the cost of a healthy diet/recommended diet in Bangladesh were identified through literature review and expert opinions [20,21,22,23,24,25,26,27,28,29,30,31,32,33,34]. These uncertainties include climate change, agricultural productivity, poverty, income inequality, affordability/accessibility, market dynamics, price volatility, awareness, inadequate policy implementation, etc.

Agricultural productivity: Bangladesh faces increasing climate variability, affecting agrarian yields and rising food prices. Erratic rainfall patterns and extreme weather events disrupt crop production, leading to uncertainty about food availability and affordability. For example, rising temperatures and rain shortfall can induce drought, reducing crop yields like wheat, corn, and rice. Moreover, warmer temperatures can create more favorable conditions for pests and diseases, increasing crop damage and low yield [23]. Changes in precipitation patterns and increased evaporation rates can exacerbate water scarcity in many regions, impacting irrigation systems and reducing water availability for agriculture [24]. These fluctuations directly influence the affordability and accessibility of nutritious foods for vulnerable populations, particularly in rural areas where food production is heavily dependent on climate-sensitive agriculture.

Climate resilience: Climate change in recent years has become inevitable to some extent, which might substantially affect agricultural productivity, leading to a limited food supply. The limited availability of different food products will, in turn, raise food prices and costs for a healthy diet, pushing a portion of the population to be unable to acquire a healthy diet [25,26]. Thus, uncertainty in Bangladesh’s resilience to climate change is an important factor in the cost of a healthy diet [25,26]. Adequate resilience to climate change will contribute to keeping agricultural production smooth and the affordable price of a healthy diet.

Market dynamics and food price volatility: Market dynamics and price volatility in Bangladesh are influenced by a combination of factors, including supply and demand dynamics, agricultural production, trade policies, weather conditions, input costs, and macroeconomic factors. Uncertainty in market dynamics directly impacts the affordability of nutritious foods for consumers by fluctuating the prices of food items and their supply. Bangladesh faces economic challenges, including inflation. Over the last year, essential commodity prices have surged by 30–62% [27]. This price hike disproportionately impacts households with limited income, making it harder to afford nutritious and diversified foods.

Affordability and accessibility: Another uncertainty lies in the socioeconomic factors affecting household purchasing power, including income inequality, unemployment rates, and inflation. These factors can constrain families’ ability to afford diverse and nutrient-rich foods, leading to a reliance on cheaper, less nutritious alternatives. In Bangladesh, about 43% of households cannot afford this cost, with rural areas bearing a heavier burden [19]. Persistent income inequality and widespread poverty limit access to nutritious foods for many Bangladeshis, especially in rural areas. Income fluctuation further exacerbates food insecurity, making it challenging for vulnerable populations to afford a diverse and healthy diet.

Equity and unemployment: An uncertainty of the cost and affordability of a healthy diet remains due to the inequity of income and the unemployment rate in Bangladesh. Bangladesh’s highly competitive job markets might leave a significant percentage of the young population unemployed and unable to afford a healthy diet. Moreover, inequity in income can raise the cost of a healthy diet as there will be a part of the population with affluence, while others will be with insolvency [22]. Thus, fluctuations in unemployment rate and equity in income inequality further exacerbate food insecurity, making it difficult for disadvantaged groups to pay for a varied and healthy diet.

Changes in dietary pattern: The dietary patterns of the people of Bangladesh have changed over the last decades [28,29]. Different fast food and ready-to-eat food intake is increasing, especially in urban areas. Developing healthy dietary patterns in Bangladesh will help people have adequate nutrient intake and reduce malnutrition. Thus, the uncertainty of the dietary pattern, whether a healthy or unhealthy pattern, will influence the cost of a healthy diet and reduce the different parameters of malnutrition prevalence.

Policy implementation and governance: Effective implementation of policies aimed at promoting food security and nutrition remains uncertain in Bangladesh. Inconsistent governance, bureaucratic hurdles, and a lack of accountability hinder efforts to address the root causes of food insecurity and ensure access to affordable, nutritious foods for all. Furthermore, the lack of infrastructure and distribution networks in remote regions hampers the availability of fresh produce and drives up transportation costs, further exacerbating food insecurity. Additionally, rapid urbanization and changing dietary preferences contribute to shifts in food demand and consumption patterns, presenting challenges for policymakers aiming to promote healthy eating habits.

Subsidies and support programs: Bangladesh’s government subsidies and assistance programs reduce the cost of basic food items, improving accessibility for low-income households [30,31]. Programs like Vulnerable Group Feeding (VGF) and Open Market Sales (OMS) aim to combat food insecurity, while subsidies on staples like rice and wheat ease financial burdens [30]. However, these initiatives prioritize calories over nutritional quality, limiting access to nutrient-rich foods like fruits and proteins. Corruption, inefficiencies, and urban-rural disparities further undermine their effectiveness. Strengthening and diversifying these programs is crucial to address affordability and nutritional adequacy.

Supply chain inefficiencies: Supply chain inefficiencies in Bangladesh significantly increase the cost of a healthy diet by limiting the availability and raising the prices of perishable and nutrient-rich foods like fruits, vegetables, and dairy. Poor infrastructure, inadequate storage facilities, and inefficient transportation systems lead to high food wastage and inconsistent supply, especially in rural areas [32,33,34]. These challenges inflate costs and reduce consumer affordability, disproportionately affecting low-income households. Addressing these inefficiencies is essential to ensure a steady supply of affordable, nutritious foods nationwide.

Seasonal and location variabilities: Seasonal and location variabilities significantly influence the cost of a healthy diet in Bangladesh [20,21]. Seasonal changes affect the availability and price of perishable foods like fruits and vegetables, often making them more expensive during the off-season. Rural areas may face lower food prices due to proximity to agricultural production, but often lack access to diverse and nutrient-rich foods. In contrast, urban areas typically have better food variety but higher prices due to transportation and storage costs. These disparities challenge the ability to ensure consistent access to affordable, healthy diets for all populations.

### 3.2. Different Scenarios for the Cost of a Healthy Diet

Expert opinions were gathered on the level of uncertainty and the significance of the identified uncertainties (Figure 1). Two uncertainties were identified from the previously discussed uncertainties based on their degree of uncertainty and degree of significance.

Limited access to nutritious foods, low purchasing power among low-income households, and disparities in food availability contribute to challenges in affording a healthy diet for many Bangladeshis. According to several studies, nutrient-dense foods cost much more than foods high in calories but low in micronutrients [19,25]. These increased prices of healthy foods are leading people to consume unhealthy foods, impacting their health significantly and resulting in child malnutrition. A recent study showed that a 5 percent increase in the price increases the risk of wasting by 9 percent and severe wasting by 14 percent [35]. To tackle malnutrition, it has been imperative to take policies to make the prices of healthy foods affordable and accessible for all. Thus, the food price volatility and changes in dietary patterns were two identified uncertainties based on their degree of uncertainty and significance. Based on the two most uncertain and most important uncertainties of the cost of a healthy diet, the following four scenarios have been developed in a 2 × 2 matrix (Figure 2).

Scenario 1: Optimal Health for All: Affordable Nutrient-Rich Diets: Scenario 1 would be the most desired, as people would have a highly healthy eating pattern with lower prices. This means that the prices of nutrient-rich food would be more affordable for the population, leading them to have more nutritious food habits with diversified foods from different food groups. This scenario would reduce diet-related diseases, resulting in more nationwide equity. In addition, households with lower incomes will be able to afford healthy food items, leading to reduced food insecurity with reduced malnutrition.

Scenario 2: High price for healthy eating (high costs, healthy eating): Scenario 2, with a highly healthy eating pattern and high food prices, will have positive and negative aspects. In this scenario, the high healthy eating pattern implies that people are consuming nutrient-rich healthy foods and making informed choices to consume fresh fruits and vegetables, protein, and whole grains, and prioritizing their healthy food habits. On the other hand, high food prices can create barriers to accessing nutritious foods, particularly for lower-income households, exacerbating food insecurity and health disparities.

Scenario 3: Crisis of Nutrition: Unaffordable and Unhealthy Diets: This scenario would be the worst, as people would have unhealthy and low-nutrient-dense eating patterns due to high food prices. High food prices can limit the intake of nutrient-rich foods, and the diets would lack essential vitamins and minerals, leading to an increased risk of malnutrition and diet-related health diseases. This would strain households, forcing them to cut their food expenditure and consume calorie-dense items. This would greatly hamper the productivity of individuals and increase health costs. Increased food prices may exacerbate food insecurity and disparities in health outcomes, with marginalized communities facing greater challenges in accessing nutritious diets.

Scenario 4: Unhealthy Eating Patterns Despite Affordable Food Prices: In this scenario, where food prices are low, people often correlate the high availability of processed foods high in sugars, fats, salt, and refined carbohydrates but low in essential nutrients. This can contribute to a diet lacking vitamins, minerals, and fiber, leading to nutritional deficiencies and related health problems. When unhealthy foods become more financially accessible, people may make cost-saving choices and choose not to spend on nutrient-rich food items. This may not only affect their immediate health outcomes, but it can also drive-up health-related costs in the long term.

### 3.3. Implications of Scenarios for Different Food System Actors

Different food actors, such as consumers, farmers, retailers, and policymakers, may have different consequences for each scenario.

#### 3.3.1. Scenario 1: Optimal Health for All: Affordable Nutrient-Rich Diets

Consumers: Affordable access to nutritious foods can improve overall health, reducing the prevalence of diet-related diseases such as obesity, diabetes, and cardiovascular conditions. At the same time, lower food prices can make healthy foods more accessible to a broader range of people, reducing food insecurity and hunger. Consumers may also save money on food purchases, freeing up resources for other necessities or savings.

Farmers and Producers: Higher consumer demand for fruits, vegetables, whole grains, and other healthy foods can increase production and sales, which will financially benefit the farmers. To meet the high demand for healthy foods and keep prices at a low margin, farmers may also need to adopt more efficient and sustainable agricultural practices. In addition, this scenario assumes support mechanisms to protect farmer incomes while keeping consumer prices low, such as targeted subsidies, improved market linkages, reduced post-harvest losses, and lower transaction costs. These measures help keep healthy foods affordable without lowering farm-gate prices, ensuring farmers receive fair earnings.

Retailers: Increased demand for healthy foods can lead to higher sales volumes for retailers, even if profit margins per unit are lower. They may need to include more healthy foods according to consumer demand, ensuring a steady supply chain of affordable healthy foods.

Policymakers: Policies that support low prices for healthy foods can lead to widespread improvements in public health, reducing the burden on healthcare systems. To ensure affordability does not negatively impact food quality or farmer livelihoods, policymakers may need to adopt approaches such as subsidies, incentives for diversified production, investments in storage and distribution, and regulations that enhance efficiency across the value chain. This requires careful monitoring and regulation to maintain healthy eating habits while supporting all actors in the food system.

#### 3.3.2. Scenario 2: High Price for Healthy Eating (High Costs, Healthy Eating)

Consumers: High food prices can place a significant financial burden on consumers, making it challenging for low- and middle-income families to afford a nutritious diet. As a result, they may need to prioritize their spending, potentially reducing expenditures on other necessities to afford healthy foods.

Farmers: High prices for healthy foods can increase revenue and profitability for farmers and producers of nutritious crops. They may invest more in quality and sustainable farming practices to maintain high standards and quality, which might require significant investments in technology and infrastructure.

Retailers: High prices for healthy foods can create opportunities for premium and specialty food markets, attracting consumers willing to pay more for quality. Retailers that offer high-quality, healthy foods may enhance their brand reputation and loyalty among health-conscious consumers.

Policymakers: Governments might need to intervene to ensure that healthy foods remain accessible, potentially through subsidies, food assistance programs, or price controls. As food prices will be higher, a large portion of the population may be unable to afford them. The government might necessitate public health campaigns to educate consumers on budget-friendly ways to maintain a healthy diet.

#### 3.3.3. Scenario 3: Crisis of Nutrition: Unaffordable and Unhealthy Diets

Consumers: High food prices will make it difficult for a large population to afford healthy diets, leading to increased health disparities and unequal health outcomes. Due to low healthy eating patterns, people are more inclined to eat unhealthy foods, making their health susceptible to health-related diseases.

Farmers: High food prices will benefit the farmers financially. However, due to the high prices of the foods, people would shift towards more processed and commercialized foods, ultimately impacting the demand for nutritious foods.

Retailers: As the market prices of nutritious foods are higher, food industries will be beneficial as the demand for processed foods will go higher, and they will increase their sales with better profit margins.

Policymakers: The government may need to take essential measures to bring down the prices of food items and give support to lower-income households. They may also arrange nutrition campaigns and road shows to educate people about healthy eating patterns and help them make informed food choices.

#### 3.3.4. Scenario 4: Unhealthy Eating Patterns Despite Affordable Food Prices

Consumers: Low food prices may encourage the consumption of cheap, processed, and nutritionally poor foods, leading to widespread nutritional deficiencies. At the same time, diets high in unhealthy, low-cost foods can also result in higher rates of obesity, diabetes, heart disease, and other chronic illnesses. Consumers may think they can save money due to low food prices in the short term, but they may face higher long-term healthcare costs and diet-related health issues.

Farmers: Farmers may prioritize growing high-yield, low-cost crops, potentially leading to a decline in the diversity and quality of agricultural production. Farmers might resort to intensive farming practices to remain profitable, which can have negative environmental impacts.

Retailers: Retailers might focus on stocking cheaper, processed foods to meet consumer demand, potentially reducing the availability of healthier options. They may sell high volumes of low-cost, unhealthy foods, but profit margins per unit may be small, affecting their overall profitability.

Policymakers: Low healthy eating patterns contribute to public health issues, increasing the burden on healthcare systems and public health resources. Policymakers may need to introduce regulations, subsidies, or educational campaigns to promote healthier eating habits and make nutritious foods more affordable.

## 4. Discussion

This study highlights that the high cost and limited affordability of healthy diets remain critical barriers to addressing malnutrition in Bangladesh. Despite progress in reducing child stunting, underweight, and micronutrient deficiencies, approximately 43% of households are unable to afford a nutrient-rich diet, with rural populations disproportionately affected. The scenario analysis provides a structured understanding of how key uncertainties—particularly food price volatility and changes in dietary patterns—can influence the cost, accessibility, and consumption of healthy diets. By exploring four plausible scenarios, the study illustrates how variations in food prices and consumer behavior can lead to markedly different nutritional outcomes, emphasizing the relevance of affordability in promoting population health.

Scenario 1, “Optimal Health for All,” demonstrates that when nutritious foods are affordable and consumption patterns are healthy, households can achieve adequate nutrient intake, reduce malnutrition, and improve equity in dietary access. Scenario 2, “High Price for Healthy Eating,” shows that even with healthy dietary choices, high food prices can limit accessibility for low-income households, highlighting the critical role of price stability in promoting nutrition. Scenario 3, “Crisis of Nutrition,” underscores the compounded risks when high prices coincide with unhealthy dietary patterns, leading to greater food insecurity and increased diet-related health burdens. Scenario 4, “Unhealthy Eating Patterns Despite Affordable Food Prices,” indicates that low prices alone do not guarantee adequate nutrition, as behavioral and cultural factors can influence unhealthy consumption patterns. Collectively, these scenarios demonstrate that addressing malnutrition requires interventions that target both affordability and behavior, rather than focusing on one dimension alone.

These findings align with previous research showing that nutrient-dense foods such as fruits, vegetables, and animal-source foods are consistently more expensive than calorie-dense but nutrient-poor alternatives [19,21,25]. Even small increases in food prices are associated with higher risks of wasting and severe wasting among children [30]. Our results also reinforce evidence that lower-income households consume fewer fruits and vegetables, indicating persistent inequities in dietary quality despite overall improvements in income and poverty reduction [11,18].

The scenario analysis provides practical insights for different food system actors. For consumers, it highlights the need for access to affordable, nutrient-rich foods and the promotion of healthier dietary patterns. For farmers and producers, it emphasizes the importance of diversifying crops and adopting sustainable practices to meet rising demand for nutritious foods. Retailers and policymakers can use scenario insights to design strategies that stabilize prices, improve supply chains, and ensure equitable access to healthy diets. This multi-actor perspective underscores the interconnectedness of economic, environmental, and behavioral factors in shaping dietary outcomes.

The foresight approach used in the present study provided forward-looking insights into the future cost and intake scenarios of healthy diets in Bangladesh while addressing the complexity and uncertainty of food system transformation. Unlike conventional analyses based only on quantitative data or short-term projections, this qualitative and participatory process enabled a more holistic exploration of possible futures. It allowed diverse stakeholders to engage in collective reflection, co-create future visions, and identify leverage points for improving diet affordability and nutrition outcomes. The scenario-building process encouraged participants to reveal both risks and opportunities under alternative policy and socioeconomic conditions. By linking qualitative insights with strategic decision-making, this foresight approach fostered dialogue between science, policy, and practice and contributed to anticipatory governance for food system transformation in Bangladesh and similar contexts.

A key strength of this study is the integration of secondary data, literature review, and expert consultation to assess uncertainties affecting diet cost, combined with scenario analysis to explore plausible futures. However, limitations include reliance on national-level data that may not fully capture regional or intra-household variations, and the subjective nature of expert evaluations in scenario development. Despite these limitations, the study provides actionable evidence for policymakers, researchers, and stakeholders seeking to improve dietary quality and reduce malnutrition in Bangladesh.

Future research should examine individual-level dietary intake, evaluate the affordability of nutrient-rich diets across different demographic groups, and assess the effectiveness of policy interventions. Longitudinal studies tracking the effects of food price fluctuations, climate variability, and dietary transitions on nutrition can further inform evidence-based strategies. Integrating behavioral interventions, market stabilization, and climate-resilient practices will promote equitable access to nutrient-rich diets and strengthen public health outcomes.

### Policy Recommendations

Addressing the prevailing uncertainties in the cost and accessibility of healthy diets in Bangladesh, like other underdeveloped countries, requires a coordinated set of priority actions targeting the root causes of food insecurity and malnutrition. As a first priority, policies should focus on poverty reduction, income equity, and strengthening social protection, which directly influence households’ ability to afford nutritious foods. Parallel investments in agricultural development and climate resilience are also essential to ensure a stable and diverse supply of nutrient-rich foods.

Region-specific interventions are essential, as different agroecological zones require tailored strategies to increase the availability of diverse, nutrient-rich foods. Improving nutrition knowledge and awareness through multi-sectoral education, behavior change communication, and capacity building is critical to promote healthier dietary patterns and diversify rice-based diets.

Leveraging indigenous and locally available nutrient-dense foods can help fill micronutrient gaps, and these findings should be widely disseminated. Strengthening rural infrastructure—such as roads, supply chains, cold storage, and warehouses—should be treated as a basic investment. Better infrastructure helps reduce post-harvest losses, improves market access, and lowers production and transport costs. These improvements also create a supportive environment for farmers to adopt and expand climate-resilient and innovative technologies, including vertical farming.

Coordinated action among government agencies, private sector stakeholders, farmers, and civil society is needed to stabilize food prices, enhance market efficiency, and promote sustainable production practices. Government ministries and local authorities might play the main role in ensuring policy coordination, while the private sector and farmers’ groups help improve markets and support the use of new technologies. Finally, strong multilevel governance and policy coherence at national and subnational levels are essential to create an enabling environment for equitable access to affordable, nutritious diets.

## 5. Conclusions

This study demonstrates that the high cost and limited affordability of healthy diets remain major barriers to addressing malnutrition in Bangladesh. Despite improvements in income and reductions in poverty, many households, particularly in rural areas, still cannot access nutrient-rich diets. Key uncertainties, including food price volatility and changing dietary patterns, further challenge dietary adequacy. Scenario analysis indicates that Scenario 1 (Optimal Health for All: Affordable Nutrient-Rich Diets) emerged as the most effective and viable scenario for reducing malnutrition. This scenario represents a future in which nutritious diets become both accessible and affordable through strengthened market systems, coherent policies, and improvements across the food value chain. Achieving this desirable scenario will require coordinated actions among consumers, farmers, retailers, and policymakers. Addressing market inefficiencies, supporting sustainable food production, and implementing policies that enhance access to nutritious foods are essential. Strengthening government programs and promoting healthy dietary practices can mitigate risks from economic and environmental shocks. Reducing the cost of a healthy diet is, therefore, critical not only for improving individual food choices but also for advancing public health, lowering malnutrition, and fostering resilient, healthier communities in Bangladesh.

## Figures and Tables

**Figure 1 foods-14-04237-f001:**
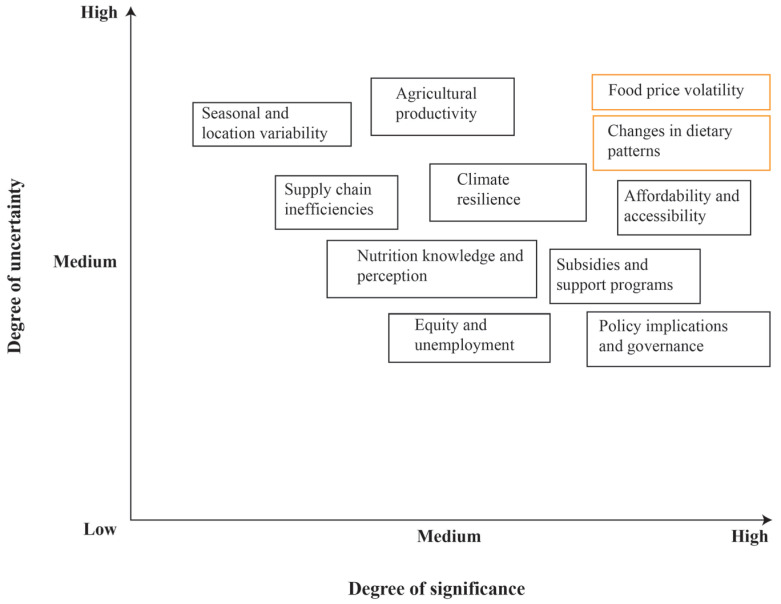
Uncertainties based on their degree of uncertainty and degree of significance.

**Figure 2 foods-14-04237-f002:**
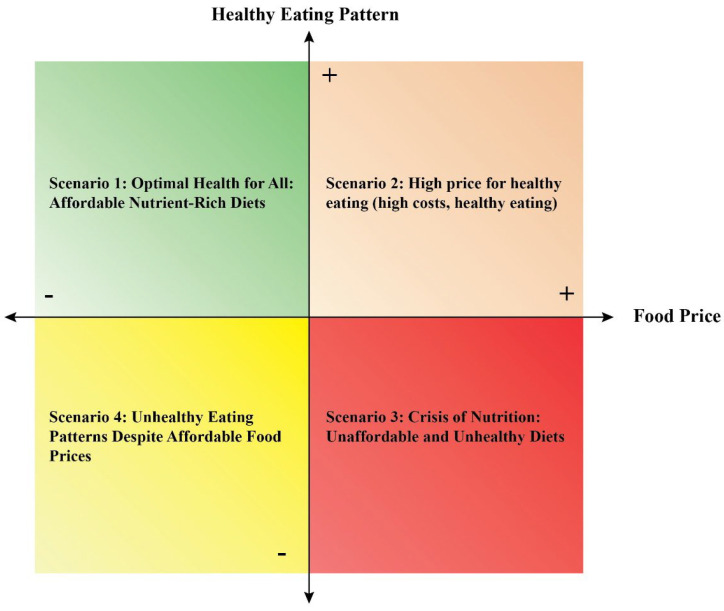
Different scenarios of the cost of a healthy diet obtained from the 2 × 2 matrix of the uncertainties of the cost of a healthy diet.

## Data Availability

The original contributions presented in this study are included in the article. Further inquiries can be directed to the corresponding author.

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
