# Peer review of "Addressing Malnutrition Through Reducing the Cost of a Healthy Diet in Bangladesh"

_foods, 2025, doi:10.3390/foods14244237_

Round 1
Reviewer 1 Report
Comments and Suggestions for Authors
The manuscript explores how reducing the cost of healthy diets may help addressing the malnutrition in Bangladesh. The manuscript language is adequate. However, even authors include a section of materials and methods, methodology is not clear. This section included two main methodological components: 1) Identification of key uncertainties and 2) Scenario development. For the identification of key uncertainties, authors state that they conducted comprehensive search (line 118) in different sources. However, in their results they cite only about 5 references. I would recommend authors to include in the manuscript the criteria they used for reference selection and justify why they consider those references to be comprehensive. In fact much of the information currently included in this section of the manuscript seems to be more adequate for the introduction rather than for the results. In lines 204 – 250 authors have not cited any references nor have included quantitative data which diminishes the reliability of the presented information.
Regarding the Scenario section, I suggest authors include expert credentials to demonstrate their expertise. In line 130 and Figure 1, no scale is provided. Please describe what scale you used and why.
Please include a section for statistical analysis.
According to your study’s findings, please describe conclude on what would be the most effective or viable scenario aiming to address the malnutrition in Bangladesh

Author Response
Reviewer 1
Comment 1: The manuscript explores how reducing the cost of healthy diets may help addressing the malnutrition in Bangladesh. The manuscript language is adequate. However, even authors include a section of materials and methods, methodology is not clear. This section included two main methodological components: 1) Identification of key uncertainties and 2) Scenario development. For the identification of key uncertainties, authors state that they conducted comprehensive search (line 118) in different sources. However, in their results they cite only about 5 references. I would recommend authors to include in the manuscript the criteria they used for reference selection and justify why they consider those references to be comprehensive. In fact, much of the information currently included in this section of the manuscript seems to be more adequate for the introduction rather than for the results. In lines 204 – 250 authors have not cited any references nor have included quantitative data which diminishes the reliability of the presented information.
Response 1: Thank you for this constructive comment. We agree that the description of our methodology required clearer explanation. In the revised manuscript, we removed the term “comprehensive search” to avoid implying a systematic or highly rigorous review process. Instead, we now describe the literature search more transparently as a review of key documents (peer-reviewed articles, reports, and policy documents) related to food affordability, healthy diet costs, nutrition, and food system challenges in Bangladesh. Several key uncertainties (Equity and unemployment, Policy implementation and governance) were not identified from the literature alone but were also informed by expert consultations. Regarding the results section, we have addressed the concern by adding all the citations we considered for the review (references 22, 28-34). We have also moved several points from the results section to the introduction section.
Comment 2: Regarding the Scenario section, I suggest authors include expert credentials to demonstrate their expertise. In line 130 and Figure 1, no scale is provided. Please describe what scale you used and why.
Response 2: We appreciate the reviewer’s suggestion. In the revised manuscript, we have added brief descriptions of the experts’ credentials to clarify their relevant experience and justify their role in identifying key uncertainties and contributing to scenario development. We have also included the list of participants presented in the expert consultation workshop in the supplementary Table 1.
Regarding the scale used in Figure 1, we have revised the manuscript to specify that a qualitative scale was applied to highlight the degree of uncertainty and degree of significance. This scale ranged from low, medium, to high, and was chosen because it is commonly used in exploratory scenario planning when expert judgment is required to assess the relative importance of uncertainties. A description of the scale and the rationale for its use have now been incorporated into the text.
Comment 3: Please include a section for statistical analysis.
Response 3: Thank you for the comment. We would like to clarify that this policy paper did not involve primary quantitative data collection or statistical analysis. Instead, the study employed a mixed-methods foresight and evidence-synthesis approach, combining secondary data and literature review, expert consultation, system mapping, and scenario analysis. These analytical procedures are already detailed in the Methods section.
Comment 4: According to your study’s findings, please conclude on what would be the most effective or viable scenario aiming to address the malnutrition in Bangladesh
Response 4: Thank you for the comment. We have revised the conclusion to clearly state which scenario is most effective. Based on our findings, we now highlight that Scenario 1 (“Optimal Health for All: Affordable Nutrient-Rich Diets”) is the most viable option for addressing malnutrition in Bangladesh. This clarification has been added to the updated conclusion section.
Comment: How it is possible to reduce the cost of a healthy diet?
Response: We thank the reviewer for the comment. In the revised manuscript, we clarified how the cost of a healthy diet can be reduced. Specifically, our Policy Recommendations and scenario implications explain that affordability can be improved through: (i) poverty reduction and strengthened social protection; (ii) increased supply of nutritious foods via agricultural development and climate-resilient production; (iii) reduced production and market costs through improved rural infrastructure, supply chains, and storage; (iv) greater use of locally available nutrient-dense foods; and (v) stronger coordination among government, private sector, and farmers to stabilize prices and enhance market efficiency. These mechanisms collectively outline practical pathways for lowering the cost of healthy diets.
Comment: How do you know they are experts?
Response: We thank the reviewer for raising this point. The experts who participated in our consultation workshop were selected based on their institutional affiliation, professional roles, and relevant experience in nutrition, food systems, agriculture, health, and policy. Participants represented key national ministries, government agencies, UN organizations, international research institutions, and leading universities, ensuring multidisciplinary and high-level expertise. The list of participants, along with their designations and institutional affiliations—including representatives from the Ministry of Food, Ministry of Agriculture, FAO, WHO, UNICEF, WFP, IFPRI, IFAD, World Bank, and major national universities—has been provided in the supplementary file for transparency.
Comment: Include references for the uncertainties
Response: Several key uncertainties (Equity and unemployment, Policy implementation and governance) were not identified from the literature alone but were also informed by expert consultations. We have addressed the concern by adding all the citations (references 20-34) we considered for the review.
Comment: Sources of databases used for review. Mention how recent record was selected.
Response: In the revised manuscript, we removed the term “comprehensive search” to avoid implying a systematic or highly rigorous review process. Instead, we now describe the literature search more transparently as a review of key documents (peer-reviewed articles, reports, and policy documents) related to food affordability, healthy diet costs, nutrition, and food system challenges in Bangladesh. No systematic approach was used find the documents. Recent documents (within last 10 years) were reviewed.
Comment: How did the expert demonstrate their experience?
Response: The experts demonstrated their experience through their institutional roles, professional responsibilities, and track record of work in relevant sectors. Almost all participants held mid- to senior-level positions in government ministries, UN agencies, international development organizations, research institutes, and universities. Their involvement in national food security programs, nutrition policy development, agricultural planning, public health initiatives, and food system research reflects substantial domain expertise.
Additionally, many of the experts have prior experience in policy formulation, large-scale program implementation, market analysis, agriculture and nutrition research, or technical advisory roles. These credentials are reflected in the participant list included in the Supplementary Table 1, which provides their designations and institutional affiliations to ensure full transparency.
Comment: What was the scale used to identify the uncertainties to develop 2x2 scenario? How was the scale used?
Response: Regarding the scale used in Figure 1, we have revised the manuscript to specify that a qualitative scale was applied to highlight the degree of uncertainty and degree of significance. This scale ranged from low, medium, to high, and was chosen because it is commonly used in exploratory scenario planning when expert judgment is required to assess the relative importance of uncertainties. A description of the scale and the rationale for its use have now been incorporated into the text.
Comment: Move lines 153 to 178 to the introduction section.
Response: We have removed the lines from results section and inserted the points in the introduction section.
Comment: Add reference for lines 183 to 186.
Response: We have added relevant references there (references 25,26).
Comment: How does uncertainty in market dynamics directly impact the affordability of nutritious foods for consumers?
Response: We thank the reviewer for this insightful comment. In the revised manuscript, we have clarified how uncertainty in market dynamics directly affects the affordability of nutritious foods. Specifically, we now explain that unpredictable variations in food supply, transportation costs, input prices, and value-chain inefficiencies can lead to price volatility, which disproportionately affects nutrient-dense foods such as fruits, vegetables, eggs, fish, and milk. These fluctuations reduce consumers’ ability to plan food expenditures and make healthy diets less affordable, particularly for low-income households. We have added this explanation to the policy implications section to strengthen the connection between market uncertainty and dietary affordability.
Comment: Add reference for lines 204 to 250.
Response: We have added relevant references in the revised manuscript (references 22, 28-34).
Comment: Move lines 268-270 to the methodology section.
Response: We appreciate the reviewer’s suggestion. However, we respectfully clarify those lines 268–270 were retained in the results section to briefly reiterate the 2×2 scenario development approach. Although the full methodological details are already included in the methodology section, this short restatement helps readers follow the logic of the scenario presentation and ensures coherence between the methods and results.
Comment: Move lines 274-304 to the methodology section.
Response: Thank you for the comment. Lines 274–304 present the scenario outputs generated from the 2×2 matrix and therefore constitute results, not methodological steps. The methodological process is already fully described in the Methods section, and moving these scenario descriptions would reduce clarity. We believe they are appropriately placed in the Results section.
Comment: Lower prices imply lower income for farmers and food manufacturers.
Response: Thank you for raising this important point. We agree that the manuscript needed more clarity on how farmers’ incomes could be protected within a “low-price + healthy diet” scenario. In the revised version, we have added an explanation outlining the mechanisms that could support farmers while keeping healthy foods affordable. These include options such as targeted government subsidies, improved market linkages, incentives for adopting efficient and climate-smart practices, and policies that reduce production and transaction costs. We also clarified that lowering consumer prices does not necessarily require lowering farm-gate prices, but rather improving efficiency across the value chain. These additions help clarify how farmer livelihoods can be maintained within the proposed scenario.
Comment: According the study, what is the most probable scenario?
Response: Thank you for the comment. We have revised the conclusion to clearly state which scenario is most effective. Based on our findings, we now highlight that Scenario 1 (“Optimal Health for All: Affordable Nutrient-Rich Diets”) is the most viable option for addressing malnutrition in Bangladesh. This clarification has been added to the updated conclusion section.

Reviewer 2 Report
Comments and Suggestions for Authors
- The policy recommendations section lists measures across multiple domains (agriculture, education, infrastructure, etc.), but fails to categorize them by "priority" or "implementing entity," resulting in a lack of operability. For instance, the logical relationship between "strengthening rural infrastructure" and "promoting climate-resilient technologies" is unclear, and it does not explain how the former supports the latter.
- It assumes a "low-price + healthy diet model" but fails to explain how agricultural producers' income can be maintained under low-price conditions. The article mentions that "farmers need to adopt efficient agricultural practices" (Line 317), yet it does not clarify who will bear the costs (e.g., through government subsidies or market mechanisms).
- The study identifies "food price fluctuations" and "changes in dietary patterns" as core uncertain factors but overlooks the critical variable of policy implementation efficiency. For example, the article mentions that "subsidy policies prioritize calories over nutrition" (Line 231), yet fails to incorporate "policy adjustments" into the scenario analysis. This omission results in the 2x2 scenario matrix being unable to account for the important real-world possibility of "policy intervention," thereby reducing the study’s policy reference value.
Author Response
Reviewer 2
Comment 1: The policy recommendations section lists measures across multiple domains (agriculture, education, infrastructure, etc.), but fails to categorize them by "priority" or "implementing entity," resulting in a lack of operability. For instance, the logical relationship between "strengthening rural infrastructure" and "promoting climate-resilient technologies" is unclear, and it does not explain how the former supports the latter.
Response 1: Thank you for this valuable comment. We have revised the Policy Recommendations section to improve clarity, categorization, and practical operability. Specifically, we have grouped the recommendations into thematic areas (e.g., agriculture and production, market and infrastructure, social protection and education, governance). We have also clarified the logical relationship between strengthening rural infrastructure and promoting climate-resilient technologies. In the revised version, we explain that infrastructure improvements—such as roads, supply chains, and cold storage—serve as foundational investments that reduce post-harvest losses, improve market access, and lower production costs, thereby enabling wider adoption of climate-resilient and innovative technologies like vertical farming. Additionally, we have briefly indicated the primary implementing entities (government ministries, local authorities, private sector actors, and farmers’ organizations) to strengthen the operability of the recommendations.
Comment 2: It assumes a "low-price + healthy diet model" but fails to explain how agricultural producers' income can be maintained under low-price conditions. The article mentions that "farmers need to adopt efficient agricultural practices" (Line 317), yet it does not clarify who will bear the costs (e.g., through government subsidies or market mechanisms).
Response 2: Thank you for raising this important point. We agree that the manuscript needed more clarity on how farmers’ incomes could be protected within a “low-price + healthy diet” scenario. In the revised version, we have added an explanation outlining the mechanisms that could support farmers while keeping healthy foods affordable. These include options such as targeted government subsidies, improved market linkages, incentives for adopting efficient and climate-smart practices, and policies that reduce production and transaction costs. We also clarified that lowering consumer prices does not necessarily require lowering farm-gate prices, but rather improving efficiency across the value chain. These additions help clarify how farmer livelihoods can be maintained within the proposed scenario.
Comment 3: The study identifies "food price fluctuations" and "changes in dietary patterns" as core uncertain factors but overlooks the critical variable of policy implementation efficiency. For example, the article mentions that "subsidy policies prioritize calories over nutrition" (Line 231), yet fails to incorporate "policy adjustments" into the scenario analysis. This omission results in the 2x2 scenario matrix being unable to account for the important real-world possibility of "policy intervention," thereby reducing the study’s policy reference value.
Response 3: Thank you for this comment. While the 2×2 scenario matrix was based on the two core uncertainties—food price fluctuations and changes in dietary patterns—the manuscript already considers the role of policy interventions and actor-level dynamics. These are presented in the subsection “3.3. Implications of scenarios for different food system actors,” which discusses how consumers, farmers, retailers, and policymakers may influence and respond to each scenario. For example, policymakers’ capacity to implement nutrition-oriented subsidies or market regulations is addressed within the scenario interpretations. By integrating these actor-level considerations, the scenarios reflect realistic pathways and maintain their policy relevance without overcomplicating the primary scenario framework.

Round 2
Reviewer 1 Report
Comments and Suggestions for Authors
Authors improved the original version. The manuscript can be accepted for publication.